# Intravenous Administration of Mesenchymal Stem Cell-Derived Exosome Alleviates Spinal Cord Injury by Regulating Neutrophil Extracellular Trap Formation through Exosomal miR-125a-3p

**DOI:** 10.3390/ijms25042406

**Published:** 2024-02-18

**Authors:** Yutaka Morishima, Masahito Kawabori, Kazuyoshi Yamazaki, Soichiro Takamiya, Sho Yamaguchi, Yo Nakahara, Hajime Senjo, Daigo Hashimoto, Sakiko Masuda, Yoichiro Fujioka, Yusuke Ohba, Yuki Mizuno, Yuji Kuge, Miki Fujimura

**Affiliations:** 1Department of Neurosurgery, Graduate School of Medicine, Hokkaido University, Sapporo 060-8638, Hokkaido, Japan; yutaka1209@gmail.com (Y.M.); mt.kazu71@gmail.com (K.Y.); soichiro.tkmy@gmail.com (S.T.); fujimur@med.hokudai.ac.jp (M.F.); 2Regenerative Medicine and Cell Therapy Laboratories, Kaneka Corporation, Kobe 650-0047, Hyogo, Japan; 3Department of Hematology, Faculty of Medicine, Graduate School of Medicine, Hokkaido University, Sapporo 060-8638, Hokkaido, Japan; 4Department of Medical Laboratory Science, Faculty of Health Sciences, Hokkaido University, Sapporo 060-0812, Hokkaido, Japan; sakikomasuda@hs.hokudai.ac.jp; 5Department of Cell Physiology, Faculty of Medicine, Hokkaido University, Sapporo 060-8638, Hokkaido, Japan; 6Central Institute of Isotope Science, Hokkaido University, Sapporo 060-0815, Hokkaido, Japan; mizuno@ric.hokudai.ac.jp (Y.M.);

**Keywords:** spinal cord injury, mesenchymal stem cell, neutrophil, NETs, miR-125a-3p

## Abstract

Spinal cord injury (SCI) leads to devastating sequelae, demanding effective treatments. Recent advancements have unveiled the role of neutrophil extracellular traps (NETs) produced by infiltrated neutrophils in exacerbating secondary inflammation after SCI, making it a potential target for treatment intervention. Previous research has established that intravenous administration of stem cell-derived exosomes can mitigate injuries. While stem cell-derived exosomes have demonstrated the ability to modulate microglial reactions and enhance blood–brain barrier integrity, their impact on neutrophil deactivation, especially in the context of NETs, remains poorly understood. This study aims to investigate the effects of intravenous administration of MSC-derived exosomes, with a specific focus on NET formation, and to elucidate the associated molecular mechanisms. Exosomes were isolated from the cell supernatants of amnion-derived mesenchymal stem cells using the ultracentrifugation method. Spinal cord injuries were induced in Sprague-Dawley rats (9 weeks old) using a clip injury model, and 100 μg of exosomes in 1 mL of PBS or PBS alone were intravenously administered 24 h post-injury. Motor function was assessed serially for up to 28 days following the injury. On Day 3 and Day 28, spinal cord specimens were analyzed to evaluate the extent of injury and the formation of NETs. Flow cytometry was employed to examine the formation of circulating neutrophil NETs. Exogenous miRNA was electroporated into neutrophil to evaluate the effect of inflammatory NET formation. Finally, the biodistribution of exosomes was assessed using ^64^Cu-labeled exosomes in animal positron emission tomography (PET). Rats treated with exosomes exhibited a substantial improvement in motor function recovery and a reduction in injury size. Notably, there was a significant decrease in neutrophil infiltration and NET formation within the spinal cord, as well as a reduction in neutrophils forming NETs in the circulation. In vitro investigations indicated that exosomes accumulated in the vicinity of the nuclei of activated neutrophils, and neutrophils electroporated with the miR-125a-3p mimic exhibited a significantly diminished NET formation, while miR-125a-3p inhibitor reversed the effect. PET studies revealed that, although the majority of the transplanted exosomes were sequestered in the liver and spleen, a notably high quantity of exosomes was detected in the damaged spinal cord when compared to normal rats. MSC-derived exosomes play a pivotal role in alleviating spinal cord injury, in part through the deactivation of NET formation via miR-125a-3p.

## 1. Introduction

Spinal cord injury (SCI) occurs due to mechanical damage to the spinal cord, resulting in various degrees of motor, sensory, and autonomic dysfunction, with an annual incidence of 40–80 per million people [1]. This condition is complex and devastating, with limited treatment options currently available. Recent evidence suggests that neutrophil extracellular traps (NETs), consisting of extracellular DNA contents released from activated neutrophils, exacerbate the local inflammatory environment and hinder neurological recovery after spinal cord injury [2,3,4]. NETs have also been reported to exaggerate neurological condition in other neurological diseases, such as stroke and traumatic brain disease [5,6,7], and controlling NET formation is considered to be an important therapeutic intervention. Our group has focused on exploring the potential therapeutic use of mesenchymal stem cells (MSCs) for spinal cord injury [8]. We, along with other investigators, have observed that MSCs, when intravenously transplanted shortly after injury, effectively restore motor function in the animal SCI model. Nevertheless, the administered cells predominantly accumulate in the lungs and liver, with no detectable presence in the injured spinal cord [9,10,11]. This can be attributed to the bystander effect of stem cells, in which trophic factors, cytokines, and exosomes released from the cells play a role in reducing both local and systemic inflammation, consequently contributing to the rescue of the damaged spinal cord [12]. Among these approaches, stem cell-derived exosomes have been attracting considerable attention. Exosomes, nano-sized vesicles ranging from 40 to 200 nm, are characterized by a double lipid-layer membrane and encapsulate a myriad of molecules, including DNA, mRNA, microRNA (miRNA), and proteins [13]. These molecules can be transferred into target cells, offering therapeutic benefits in mitigating neurological diseases [14,15,16]. Their remarkable cryopreservation capacity and low immunogenicity have prompted considerations of MSC-derived exosomes as a promising alternative to MSC. Previous reports have demonstrated that MSC-derived exosomes exert their effects on the spinal cord by inducing a shift in the phenotype of macrophages and microglia from a proinflammatory (M1) state to an anti-inflammatory (M2) state. They also inhibit astrocytic gliosis and promote angiogenesis and neurogenesis [17,18,19,20,21,22]. However, the influence of exosomes on neutrophils, the most abundant and earliest cells recruited from the bloodstream to the spinal cord after an injury resulting in aggravation of inflammation, has not been extensively studied.

In light of these circumstances, our research aims to investigate the effects of intravenous administration of amnion-derived MSC (AMSC) exosome on SCI, with a specific focus on their interaction with neutrophils and the formation of NETs.

## 2. Results

### 2.1. Exosome Characterization

Transmission electron microscopy (TEM) images revealed that the exosomes exhibited a consistent shape with an average diameter of approximately 80 nm (Figure 1A), which was further corroborated by the size distribution data from the nanoparticle analyzer (Figure 1B). Western blot analysis demonstrated elevated expression levels of CD9 and CD63, specific markers for exosomes, in comparison to the AMSC suspension. Conversely, the expression of calnexin, a marker with low expression in exosomes, was diminished in the exosome (Figure 1C). These results confirm the presence of exosomes in the AMSC-derived samples.

### 2.2. Intravenous Exosome Administration Ameliorates SCI Damage in Mice

The animals that received exosomes exhibited significantly better neurological recovery compared to those that received phosphate-buffered saline (PBS) (Figure 2A), starting one week after administration. Kluver–Barrera staining revealed that the damaged volume of the spinal cord tissue collected at 4 weeks post-SCI was significantly lower in the exosome-treated group (*p* = 0.006) (Figure 2B,C). In the peri-injury area at 4 weeks post-SCI, there was a notable reduction in MPO-positive neutrophil infiltration in the exosome-treated group (*p* = 0.037) (Figure 3A,B). Subsequently, early neutrophil infiltration and NET formation were examined in the day 3 section. Immunofluorescence imaging revealed that CitH3-positive cells, as well as MPO-positive cells, were significantly reduced in the exosome-treated group (Figure 4A,B) (*p* = 0.023 *p* = 0.047, respectively). This phenomenon is further confirmed by the decrease in NETs, as demonstrated by peri-nuclear histone integration in the exosome-treated group (*p* = 0.003, Figure 4C,D).

### 2.3. Ex Vivo Analysis of Exosomes for NET Formation

Peripheral blood was obtained to assess NET formation in neutrophils. Initially, we examined the uptake of exosomes by neutrophils. Labeled exosomes were co-incubated with peripheral neutrophils, and it was observed that the exosomes began to accumulate around the nucleus of neutrophils approximately 10 min after co-incubation (Appendix A). Subsequently, neutrophils were activated by LPS and co-incubated with exosomes. The number of neutrophils positive for CitH3 and MPO was significantly reduced in the exosome-treated group compared to the PBS group (Figure 5A,B). This finding was further validated through flow cytometry (FACS) analysis, which demonstrated a significant decrease in CitH3-positive Ly6G+ neutrophils following exosome administration (*p* = 0.002) (Figure 5C,D).

To elucidate the functional mechanisms of exosomes in NET formation, neutrophils were electroporated with miR-125a-3p mimic and inhibitor to assess NET formation following lipopolysaccharide (LPS) activation. Exosome administration effectively inhibited NET formation, and miR-125a-3p electroporation modestly enhanced this treatment effect. Furthermore, the impact of exosome administration was entirely reversed by the addition of the miR-125a-3p inhibitor, underscoring the pivotal role of miR-125a-3p encapsulated within exosomes in NET formation (Figure 6).

### 2.4. Biodistribution of Exosomes

PET imaging of ^64^Cu-labeled exosomes at one hour post injection revealed their substantial accumulation in the liver and spleen (Figure 7A). Biodistribution data obtained by tissue detection at 24 h post injection also revealed the same trend of liver and spleen accumulation (Figure 7B). Interestingly, the exosomes were found to be significantly higher in the spinal cord in the SCI model compared to the normal rat despite their limited presence (Figure 7C). This result indicates the potential for exosome accumulation at inflammatory sites.

## 3. Discussion

In the current study, we have demonstrated that intravenously transplanted AMSC-derived exosomes successfully mitigated neutrophil activation by inhibiting NET formation, resulting in the restoration of motor function. We were able to elucidate the underlying molecular mechanism responsible for this effect, which is induced by the exosomal miR-125a-3p. Biodistribution analysis using PET revealed that the majority of exosomes accumulated in the liver and spleen, with a notable quantity also observed in the injured spinal cord.

This study is built upon our previous research, in which intravenous administration of AMSCs for spinal cord injury resulted in significant functional recovery. However, it is noteworthy that AMSCs were not detected within the injured spinal cord itself [9]. Exosomes released from stem cells are considered to play a pivotal role in the therapeutic mechanisms of stem cell treatment, and numerous experiments are currently underway in the context of SCI [23,24,25]. The mechanisms can be categorized into two distinct patterns: alleviating damage and promoting neuro/angiogenesis. A majority of studies that have embraced early exosome administration focus on damage alleviation, encompassing anti-apoptosis, preservation of the blood–spinal cord barrier, and immunomodulation. While immunomodulation often involves the deactivation of macrophages, the therapeutic role of exosomes in neutrophil deactivation is rarely explored [26]. The underlying reason for this scarcity of research may be attributed to the challenges associated with ex vivo culturing of neutrophils compared to microglia/macrophages. Inflammation is known to manifest immediately following spinal cord injury, initially activating microglia within a few hours, followed by neutrophil infiltration into the injured spinal cord during the initial 1–3 days, and culminating with a peak infiltration of macrophages around one week after the initial injury. As the first non-resident cells to be recruited to the damaged spinal cord, we posit that controlling neutrophil deactivation is of paramount importance in ameliorating the cascade of inflammation in SCI [27,28,29]. In addition to the timing of administration, the number of administrations is also of significant importance. In this report, we administered exosomes for three consecutive days following SCI, which mimics the natural pattern of exosome secretion from MSCs [30]. This approach aligns with findings from Nakazaki et al., who observed that administering exosomes in fractions over three consecutive days resulted in superior functional improvement when compared to a single administration of the total exosome dose over the same three-day period [31]. Given the reported very short half-life of exosomes, multiple administrations to adequately cover the period of neutrophil activation are considered essential.

While data regarding the functional role of NET formation in SCI models remain limited, accumulating evidence substantiates the concept that NET formation exacerbates central nervous system diseases through various pathways, including direct neural damage, microglial activation, and the induction of reactive gliosis [3,4]. Feng et al. previously documented that NET formation worsens secondary injury after SCI by promoting inflammation and disrupting the blood–brain barrier (BBB), and the degradation of NETs through the use of DNase led to an enhancement in motor function following spinal cord injury [2]. Consistent with these findings, our current study observed an improvement in motor function, coinciding with the decrease in NET levels, underscoring a strong interconnection between NETs and the secondary inflammation resulting from spinal cord injury. Understanding the mechanisms by which exosomes alleviate NET formation is essential for enhancing their functionality, yet these mechanisms remain incompletely understood. Timelapse imaging has revealed that exosomes congregate around the neutrophil nucleus, suggesting a potential role in delivering active molecules to neutrophils. In our study, we selected miR-125a-3p as a therapeutic molecule for exosomes, given its well-documented involvement in inflammation and its abundant expression in AMSCs. As a result, the miR-125a-3p inhibitor successfully counteracts the NETs-suppressing effect of exosomes, confirming its role in anti-inflammatory processes. These findings are consistent with the work of Jin et al., who have exclusively reported that miR-125a-3p suppresses NETs by targeting the NF-κB signaling pathway mediated by Toll-like receptor 4 (TLR4) [32]. Kim et al. reported that mir-125a inhibits the NF-κB regulator tumor necrosis factor alpha-induced protein 3 to negatively affect NF-κB activity [33]. Additionally, mir125b, TLR4/JNK signaling has been reported to play a role in NET formation by neutrophils [34]. miR-125a-3p is also implicated in altering the phenotype of microglia by interfering with interferon regulatory factor 5 (IRF5). Knockdown of miR-125a-3p in MSC-derived exosomes resulted in increased IRF5 expression in spinal cord tissues [35]. Mir-125 is reported to show a significant correlation with inflammatory bowel disease in humans [36]. It is likely that miR-125a-3p operates through multiple mechanisms, and further evaluation is necessary to comprehensively elucidate its anti-inflammatory mechanisms. Furthermore, MSC-derived exosomes have been reported to attenuate NET formation through the inhibition of C5b-9 assembly [37]. It is conceivable that other pathways not involving miR-125a-3p may also contribute to this process.

Furthermore, our investigation revealed that exosomes are predominantly distributed in the liver and spleen, with a discernible quantity also observed in the injured spinal cord. While prior reports have documented the presence of exosomes in the spleen and damaged spinal cord, comprehensive knowledge of their distribution throughout the entire body remains incomplete [38,39], and the data presented in our study contribute significantly to this understanding. We observed that neutrophils obtained from the circulating rat blood exhibited low NET formation, which is consistent with the accumulation of exosomes in the spleen, where neutrophils are stored. The exact role of exosome accumulation in the liver remains uncertain. It could potentially be attributed to the uptake by Kupffer cells of eliminating unidentified substances. Previous reports indicate that exosomes that accumulate in the damaged spinal cord have a direct deactivating effect on microglia, ultimately leading to functional recovery [39]. The presence of exosomes in the damaged spinal cord during the acute phase, when local and systemic inflammation plays a pivotal role, warrants further investigation.

In this report, we utilized intravenous transplantation as one of the most commonly used methods. However, other transplantation routes have also been reported for exosome administration against spinal cord injury. Intraspinal injection offers the possibility of delivering a high concentration of exosomes, but the injection itself carries the risk of additional spinal cord damage [40,41]. Intranasal injection is currently gaining attention due to its minimally invasive method of delivering exosomes to the damaged spinal cord, although the absorption rate may vary depending on the patient’s condition [39,42,43]. The intrathecal injection can deliver a high amount of exosomes with a minimally invasive procedure [44].

Several limitations in our study warrant discussion. Firstly, we were unable to comprehensively analyze the content of exosomes to identify potential therapeutic candidates. This is because various elements, including proteins, mRNA, miRNA, and other unknown molecules, may contribute to the observed functional recovery, and it is quite difficult to investigate the full role of different molecules at once. Secondly, we were not able to detect the precise organ where the exosome acts against neutrophil. We assume that it would be in the spleen; however, circulating neutrophil can also be one option. Third, we did not investigate other inflammatory parameters of neutrophils, such as lesional cytokine expression. A fewer neutrophil accumulation in the exosome group at the late phase (28 days) and reduced SCI damage imply lower expression of inflammatory cytokines.

## 4. Materials and Methods

### 4.1. Ethical Approval

Animal protocols were approved by the Animal Studies Ethics Committee of the Hokkaido University Graduate School of Medicine (approval number: 17-0065). All experimental procedures were conducted in accordance with the Institutional Guidelines for Animal Experimentation and the Guidelines for Proper Conduct of Animal Experiments by the Science Council of Japan.

### 4.2. Methods

#### 4.2.1. Culturing and Isolation of AMSC Exosomes

Human AMSC vials, provided by Kaneka Corporation (Osaka, Japan), were thawed and seeded at a concentration of 2.0 × 10^5^ cells/mL in medium. The cells were incubated at 37 °C for 5 days. The medium was then replaced with serum-free cell culture medium (MEMα, 12561056, Thermo Fisher Scientific, Waltham, MA, USA) for 2 days to obtain exosome-containing medium supernatant. Differential centrifugation methods were employed for exosome isolation as previously described with minor modification [45,46]. Initially, the supernatant was subjected to a continuous centrifugation step at 2000× *g* for 10 min and 10,000× *g* for 30 min to remove dead cells and cellular debris. Subsequently, the supernatant was subjected to ultracentrifugation at 100,000× *g* for 70 min, resulting in the formation of a pellet consisting of small vesicles corresponding to exosomes. The size and morphology of the isolated exosomes were confirmed through TEM (H-7100, Hitachi, Tokyo, Japan) and nanoparticle tracking analysis (Videodrop, Myriade, Paris, France). Additionally, Western blotting was utilized to identify specific exosome surface markers, including CD 9 (1:500, 014-27763, Fujifilm Wako, Osaka, Japan), CD 63 (1:500, 012-27063, Fujifilm Wako, Osaka, Japan), and Calnexin (1:500, ab22595, Abcam, Cambridge, UK). Protein concentration of exosome diluted in PBS was evaluated by Pierce BCA Protein Assay kit (23227, Thermo Fisher Scientific, Waltham, MA, USA). 

#### 4.2.2. SCI Model and Exosome Administration

Female Sprague-Dawley rats, aged nine weeks and weighing between 200 and 233 g, were obtained from CLEA Japan, Inc., Japan. Female rats were selected because they possess shorter urinary tract than male rats, which lowers the chance of sepsis caused by anuresis after spinal cord injury. The rat spinal cord injury model was established following previously described methods [9,47]. In brief, the rats underwent dorsal laminectomy (T6–7) while under isoflurane gas anesthesia. Immediately after the laminectomy, a modified aneurysm clip (07-943-30, Mizuho, Japan) was used to create a spinal cord injury by extradural pinching of the spinal cord for 1 min [48]. Postoperative care included assisting the rats in voiding their bladders 2–3 times a day until they regained independent urination. Twenty-four hours post-spinal cord injury, the Basso–Beattie–Bresnahan (BBB) score was assessed [49], and rats with a non-zero BBB score were excluded from the study. Subsequently, 1 mL of AMSC-derived exosomes (100 μg) adjusted with PBS, or 1 mL of PBS alone, was administered via the tail vein. Exosomes were given for three consecutive days. BBB scores were then assessed at 1-week intervals for a total of 4 weeks following the surgery.

#### 4.2.3. Histological Analysis

Immunohistochemistry was conducted to assess the volume of the injured spinal cord, neutrophil infiltration, and NET formation in the spinal cord on either day 3 or day 28, as previously reported [9,47]. On the day of sacrifice, the rats were deeply anesthetized and subjected to transcardial perfusion with PBS followed by 4% paraformaldehyde (PFA). The spinal cords were then extracted, fixed in 4% PFA for 24 h, embedded in paraffin, and sliced into serial longitudinal sections measuring 10 μm in thickness using a cryostat microtome (LEICA RM2125 RTS, Leica Biosystems, Nussloch, Germany). Kluver–Barrera staining, which involves Luxol fast blue (LFB) absence, was performed to assess the volume of the injured spinal cord 28 days after SCI. The following equation was employed to calculate the lesion volume: Lesion volume (mm^2^) = πD^2^ (H1 + H2)/6, where H1 represents the lesion length from the epicenter to the rostral end, H2 denotes the lesion length from the epicenter to the caudal end, and D is the diameter of the epicenter [9,47]. Neutrophil infiltration at day 28 was evaluated using an anti-myeloperoxidase antibody (1:1000, ab208670, Abcam, Waltham, MA, USA), followed by Histofine Simple Stain MAX-PO (Nichirei Biosciences Inc., Tokyo, Japan) for 30 min and reacted with 3,3′-diaminobenzidine (DAB) (Simple Stain DAB Solution, Nichirei Biosciences Inc., Tokyo, Japan) for 3 min. Images were obtained from the peri-damaged lesion (5 mm rostral and caudal from the epicenter of the damaged lesion). A total of five non-overlapping areas were selected, and the cells exhibiting positive signals were semi-quantitatively counted using automated cell counting software https://www.keyence.com/products/microscope/fluorescence-microscope/bz-x700/models/bz-h4c/ (Hybrid Cell Count, BZ-X Analyzer, Keyence, Osaka, Japan). Immunofluorescent staining was performed to assess NETs on day 3 specimens, as previously described [50,51]. In brief, spinal cords were harvested without PFA fixation and immediately mounted in the OCT compound, and Fresh-frozen sections were cryosectioned in the longitudinal plane. The spinal cord sections were incubated overnight at 4 °C with mouse anti-Histone H3 (citrulline R2 + R8 + R17:CitH3) (1:00, Ab5103, Abcam, Waltham, MA, USA) and Anti-mouse/rat MPO-FITC (1:200, LS-C140180, LSBio, Shirley, MA, USA), followed by Alexa Fluor 594 goat anti-mouse (1:500, Thermo Fisher Scientific, Waltham, MA, USA). A mounting agent with DAPI or Hoechst was used to visualize cell nuclei. The number of MPO-positive neutrophil and CitH3-positive NETs, as well as the ratio of integrated CitH3 over Hoechst, were evaluated as previously described.

#### 4.2.4. Ex Vivo Assessment of Neutrophil and NET Formation

Neutrophils were isolated from the whole blood obtained by heart puncture of 10-week-old C57BL/6 mice according to the manufacturer’s protocol (Neutrophil Isolation KIT, Cayman Chemical, Ann Arbor, MI, USA). The quality of the isolated neutrophils was further confirmed by flow cytometry (FACS) by the expression of Ly6G (60-5931-U100, Tonbo Biosciences, Tokyo, Japan). In order to assess the NET formation in ex vivo neutrophils, isolated neutrophils were adjusted to a concentration of 5 × 10^5^ cells and incubated with a total of 400 μL of the cell suspension in a 4-well chamber slide at 37 °C for 30 min. Subsequently, exosomes at a concentration of 100 μg/25 μL or 25 μL of PBS, along with 400 μL LPS at a concentration of 100 μg/mL, were added to the cells and incubated at 37 °C for 3.5 h as previously described [52]. Following the incubation, the cells were washed and then fixed with 300 μL of fixation buffer for 15 min. Then, 300 μL of 0.1% NP-40 was added and incubated for 5 min to permeabilize the cells. Mouse anti-Histone H3 was added, followed by Alexa Fluor 594 goat anti-mouse (1:500, Invitrogen, Carlsbad, CA, USA) and anti-mouse/rat MPO-FITC. Measurement of the ratio for CitH3 positive NETs over MPO positive neutrophil was evaluated using an automated cell/area counter (BZ-X Analyzer, Keyence, Osaka, Japan) at a magnification of 10×.

To elucidate the uptake of exosomes into neutrophils, exosomes were labeled using the ExoSparkler Exosome Membrane Labeling Kit (343-09661, Dojindo, Japan) following the manufacturer’s instructions [53]. Neutrophils, adjusted to a concentration of 5 × 10^4^ cells/mL, were co-incubated with the labeled exosomes at a concentration of 100 μg/25 μL in a glass-bottom dishes (35-mm-diameter; Matsunami Glass Industry Glass, Osaka, Japan). Time-lapse fluorescence imaging and data analysis were performed essentially as described previously [54]. In brief, the cells were placed in a stage-top incubation chamber maintained at 37 °C on a Nikon Eclipse Ti2-E microscope (Nikon, Tokyo, Japan) equipped with a KINETIX22 scientific complementary metal oxide semiconductor (sCMOS) camera (Teledyne Photometrics, Tucson, AZ, USA), PlanApo 20×/0.8, or 60×/1.2 objective lenses, a TI2-CTRE microscope controller (Nikon), a TI2-S-SE-E motorized stage (Nikon). The cells were illuminated with an X-Cite turbo system (Excelitas Technologies, Waltham, MA, USA). The sets of excitation and emission filters and dichroic mirrors adopted for this observation included GFP HQ (Nikon) for ExoSparkler or DAPI-U HQ (Nikon) for Hoechst 33342. Confocal images and super-resolution images were acquired with an X-Light V3 spinning disk confocal unit (CrestOptics, Roma, Italy) and a DeepSIM (CrestOptics) for Eclipse Ti2 equipped with a Prime BSI sCMOS camera (Teledyne Photometrics), respectively. The cells were illuminated with CELESTA light engines (Lumencor, NW Greenbrier Parkway, Beaverton, OR, USA). The time-lapse imaging was configured at 5-minute intervals, covering a total duration of 1275 min.

Flow cytometry was further conducted to evaluate the NET formation in neutrophils as previously described with modification [55]. Fluorophore-conjugated or biotinylated monoclonal antibodies specific to mouse antigens were enumerated as follows: PE-Ly6G (clone 1A8, BD Pharmingen), PerCP cy5.5-CD11b (clone M1/70, eBioscience), MPO-FITC, and CitH3. The secondary reagent employed was Alexa Fluor 594 goat anti-mouse (Invitrogen, Carlsbad, CA, USA). Neutrophils were co-incubated with exosomes (100 μg/25 μL) or PBS (25 μL) for 3.5 h and subsequently activated for 24 h with LPS at a concentration of 1 μg/mL under standard conditions of 37 °C. Multiparametric analyses of the stained cell suspension were conducted utilizing a FACS Aria III cell sorter (BD) with FACS Diva software https://www.bdbiosciences.com/en-us/products/software/instrument-software/bd-facsdiva-software (BD). Neutrophils expressing NETs were delineated based on the CitH3+Ly6G+CD11b+ gating strategy and subsequently compared between the exosome-treated and PBS-treated groups.

To elucidate the therapeutic mechanism of exosomes on NET formation in neutrophils, we conducted an evaluation of exosomal miRNA. Based on a literature search and confidential data provided by Kaneka Corporation, in which miRNA-125a-3p (miR-125a-3p) were abundantly expressed in the AMSC, we selected miR-125a-3p as a candidate molecule. The electroporation of the miR-125a-3p mimic and inhibitor (has-miR-125a-3p, ThermoFisher Scientific, Waltham, MA, USA) was performed according to the manufacturer’s recommendations. Neutrophils isolated from C57BL/6 mice underwent two washes with PBS without Ca^2+^ and Mg^2+^ and were then adjusted to 2 × 10^6^ cells. After removing the supernatant, 90 μL of R buffer (Thermo Fisher Scientific, Waltham, MA, USA) was added, and the electroporation procedure was conducted using a voltage of 1720 V, a pulse width of 10 ms, and a total of 2 pulses for 20 mM of miR-125a-3p mimic and 100 mM of miR-125a-3p inhibitor. Subsequently, the cells were seeded in a 1.5 mL microtube containing 0.5 mL of pre-warmed supplemented DMEM. After 12 h, exosomes were added to the exosome group, the group electroporated with the miR-125a-3p mimic (mimic group), and the group electroporated with the miR-125a-3p inhibitor (inhibitor group). The control group was established without exosome or miR supplementation. All of the groups were then stimulated with 100 μg/mL of LPS. Following a 24-h interval, immunofluorescence staining, as previously described, was used to quantify CitH3-positive cells within the population of DAPI-positive cells.

#### 4.2.5. Biodistribution

Animal positron emission tomography/computed tomography (PET/CT) was utilized to evaluate the biodistribution of the intravenously administered exosome. The method for isotope pre-labeling of ^64^Cu was performed as previously described with minor modifications [56]. First, the solution of exosomes was replaced with 0.1 M Na_2_CO_3_ buffer (pH 10) by the PD-SpinTrap G-25 column (Cytiva, Tokyo, Japan). Separately, p-SCN-Bn-NOTA (Macrocyclics, Plano, TX, USA) was dissolved in 0.1 M acetate buffer (pH 6.0) and incubated with ^64^CuCl_2_ (138 MBq, PDRadiopharma Inc., Tokyo, Japan) at room temperature for 20 min to provide ^64^Cu-SCN-NOTA. The concentration of p-SCN-Bn-NOTA in the reaction was 190 μM (106.4 mg/L), and the radiochemical yield of this step was 89%, as determined by thin-layer chromatography (TLC). ^64^Cu-SCN-NOTA (12.5 μg) was then incubated with exosomes (526 μg) dissolved in 0.1 M Na_2_CO_3_ buffer (pH 10) at 37 °C for 3 h. Finally, the reaction mixture was purified by PD-SpinTrap G-25 column three times to remove excess ^64^Cu-SCN-NOTA and other low-molecular-weight impurities. At the time of this gel filtration, the reaction buffer was replaced with PBS. The radiochemical purity of ^64^Cu-NOTA-exosome after the purification was 90.8%, as determined by TLC. The overall radiochemical yield without decay correction was 8.1%. All TLC analyses were performed using iTLC-SG plates (Agilent Technologies, Palo Alto, CA, USA), and the developing solvent was 50 mM EDTA (pH 5.5). The radioactivity on TLC was quantified by autoradiography (Fla-7000, Fujifilm, Japan). In this TLC system, the Rf values of ^64^Cu-NOTA-exosome, ^64^Cu-SCN-NOTA, and [^64^Cu]CuCl_2_ (forming complex with EDTA during the development) were 0.0, 0.5–0.8, and 0.9–1.0, respectively. The biodistribution of ^64^Cu-NOTA-exosome was evaluated in normal and spinal cord injury model rats (9-week-old Sprague-Dawley). For spinal cord injury model rats, the experiment was performed the day after the surgery. ^64^Cu-NOTA-exosome (100 kBq, 27 μg of exosome) was injected into rats via the tail vein. The rats were sacrificed at 24 h post-injection, and the organs of interest were removed and weighed. The radioactivity of these organs was determined by a gamma counter (2480 Wizard 2 gamma counter, PerkinElmer, Waltham, MA, USA). The uptake values of ^64^Cu-NOTA-exosome are expressed as % injected dose per gram of organ (%ID/g). PET/CT imaging was performed on a spinal cord injury model rat (*n* = 1) using an Inveon preclinical small-animal multimodality PET/CT system (Siemens Medical Solutions, Knoxville, TN, USA). PET images of ^64^Cu-NOTA-exosome were acquired over 60–80 min after the administration of ^64^Cu-NOTA-exosome (8.8 MBq, 149 μg of exosome). CT images were acquired following the PET scan. During the image acquisition, the rat was anesthetized using 2.0–2.5% isoflurane. The PET and CT data were reconstructed using the Feldkamp method and 3D-OSEM (2 iterations, 18 MAP iterations), respectively. PET and CT images were analyzed using an Inveon research workplace software v. 4.1 (Siemens Medical Solutions, Knoxville, TN, USA).

#### 4.2.6. Statistical Analyses

All assessments were performed by blinded investigators. The data have been presented as mean ± standard error. Statistical analyses were performed using JMP Pro 14 software (SAS Institute Inc., Cary, NC, USA). Statistical comparisons between groups were made using the Welch’s *t*-test or Wilcoxon test. Probability values of *p* < 0.05 were considered statistically significant.

## 5. Conclusions

AMSC-derived exosomes play a crucial role in mitigating spinal cord injury, partially achieved by deactivating neutrophil NET formation via miR-125a-3p. Biodistribution analysis further indicates that the majority of exosomes are concentrated in the liver and spleen.

## Figures and Tables

**Figure 1 ijms-25-02406-f001:**
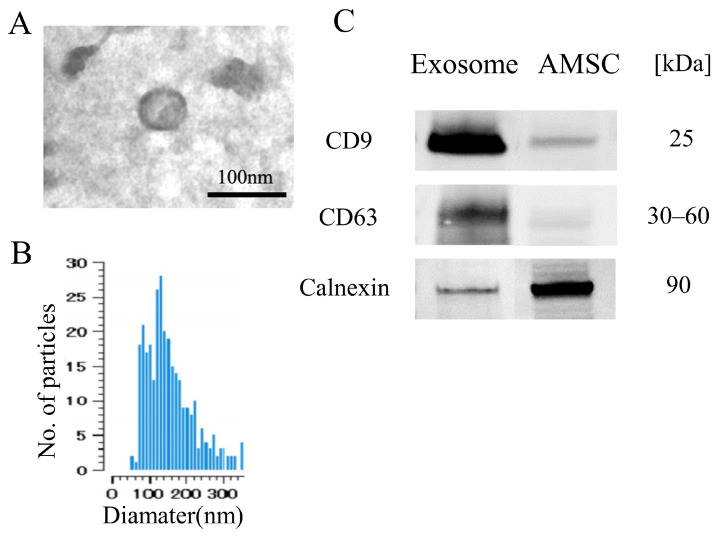
Characteristics of AMSC-derived exosome. (**A**) Electron microscopy shows vesicles with round shape morphology with diameter of approximately 100 nm. (**B**) Histogram of the size distribution by nanoparticle analyzer revealed that exosome size ranged from 70 to 300 nm with peak of 180 nm. (**C**) Western blot detection showed the exosome exhibited CD9 and CD63, while less detection was noted for Calnexin compared with the AMSC.

**Figure 2 ijms-25-02406-f002:**
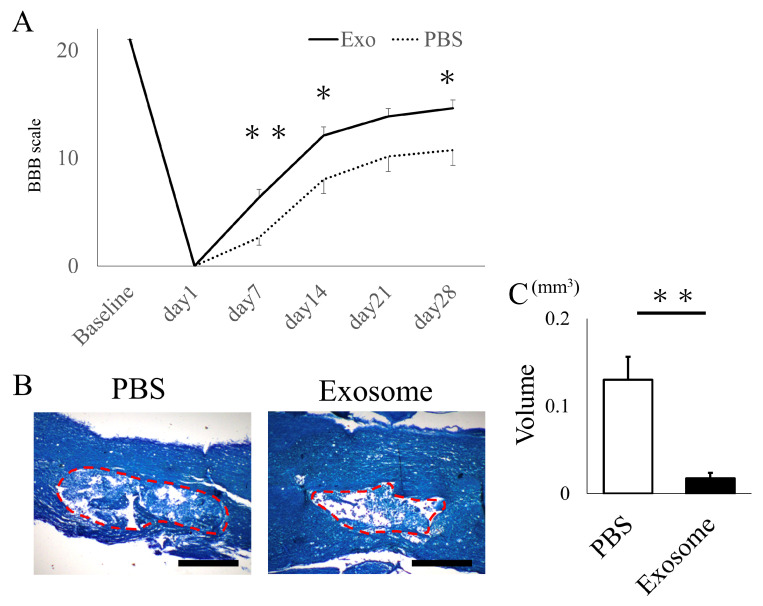
Exosome administration ameliorated neurological function and reduced damaged lesions. (**A**) Animals that received exosomes demonstrated significantly enhanced neurological recovery, starting from 1 week after transplantation, in comparison to animals that received saline. (**B**) The Kluver–Barrera staining of the spinal cord 4 weeks after SCI showed the degree of spinal cord damage (the red dotted area) (scale bar = 200 μm). (**C**) The exosome-treated group exhibited a substantially smaller damaged lesion volume compared to the PBS group (*p* = 0.0061). * *p* < 0.05, ** *p* < 0.01.

**Figure 3 ijms-25-02406-f003:**
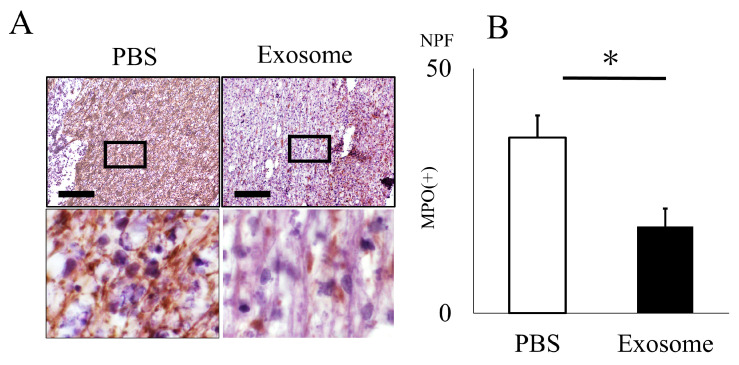
Neutrophil infiltration is inhibited by exosome administration. (**A**) Neutrophil infiltration in the spinal cord 4 weeks after SCI was assessed using the anti-MPO antibody. (**B**) The count of MPO-positive cells within the spinal cord was notably reduced in the exosome-treated group as compared to the PBS group (scale bar = 20 μm, *p* = 0.037). * *p* < 0.05.

**Figure 4 ijms-25-02406-f004:**
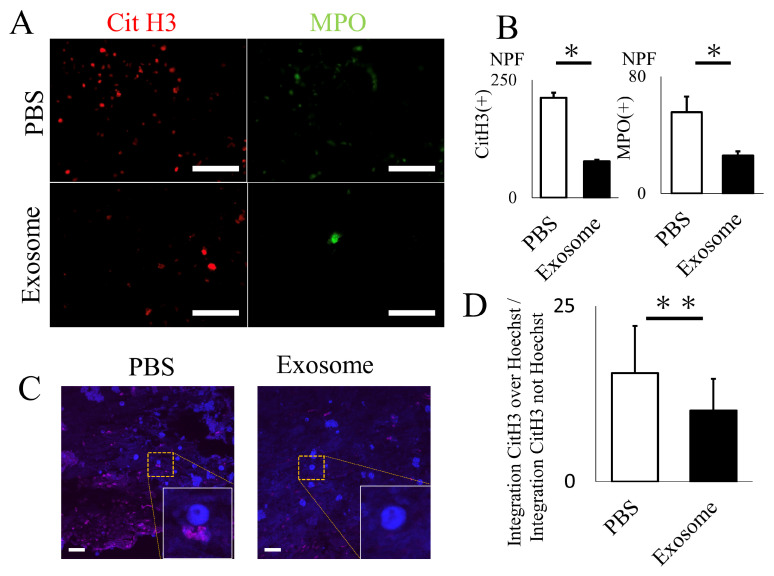
Exosome ameliorated NET formation in spinal cord. (**A**,**B**) The quantity of NETs (CitH3) and activated neutrophils (MPO), assessed via immunofluorescence, were significantly reduced in the spinal cord following exosome treatment (scale bar = 50 μm, *p* = 0.0231, *p* = 0.0466, respectively) (**C**,**D**) Existence of NETs were further evaluated by super-resolution microscopy. Note that CitH3 is extruded around the nuclear in the PBS group. (scale bar = 20 μm, *p* = 0.0027). * *p* < 0.05, ** *p* < 0.01.

**Figure 5 ijms-25-02406-f005:**
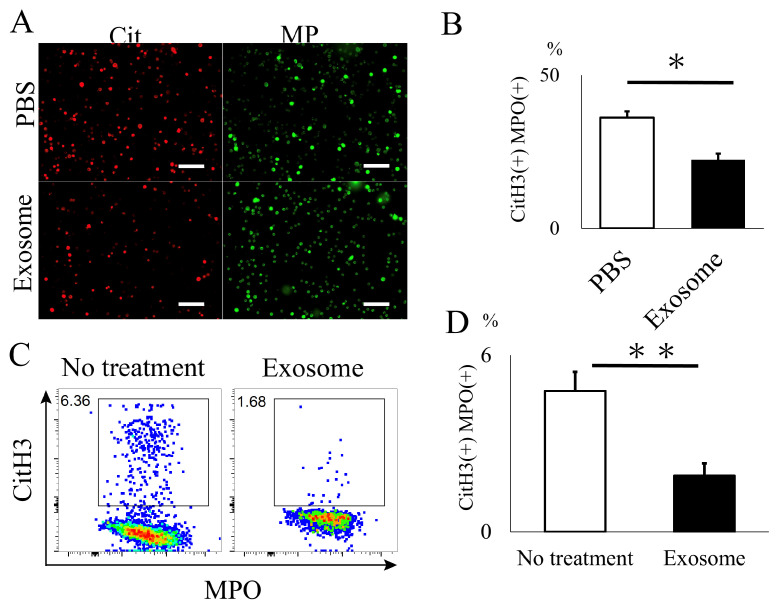
In vitro analysis of NET formation in mouse neutrophil. (**A**,**B**) CitH3 formation was significantly reduced in neutrophils treated with exosomes. (scale bar = 50 μm, *p* = 0.0101) (**C**,**D**) Flow cytometry analysis further demonstrated a decrease in cells expressing NETs in the exosome-treated group. (*p* = 0.0019). * *p* < 0.05, ** *p* < 0.01.

**Figure 6 ijms-25-02406-f006:**
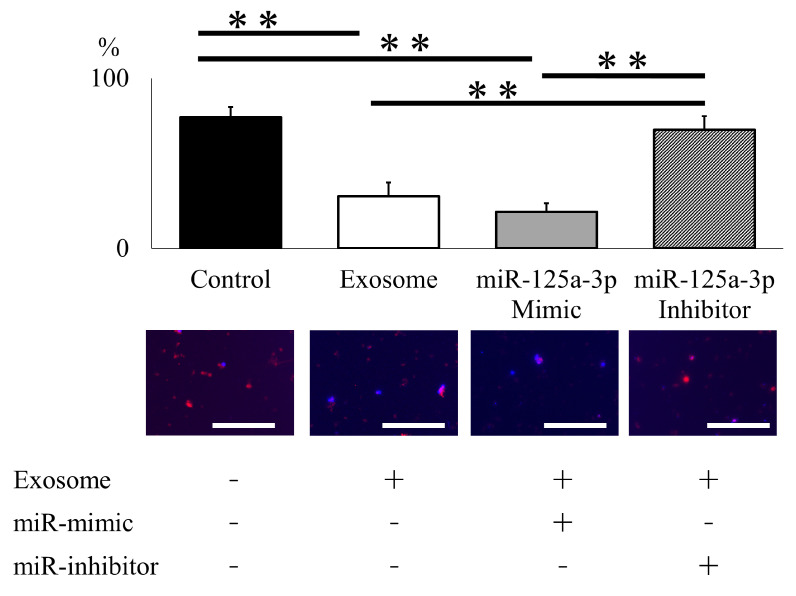
miR-125a-3p inhibits NET formation in mouse neutrophil. Exosome administration successfully attenuated NET formation following LPS administration, and miR-125a-3p mimic modestly enhanced treatment effect. The effect of exosome was entirely reversed by the addition of the miR-125a-3p inhibitor. (scale bar = 50 μm). ** *p* < 0.01.

**Figure 7 ijms-25-02406-f007:**
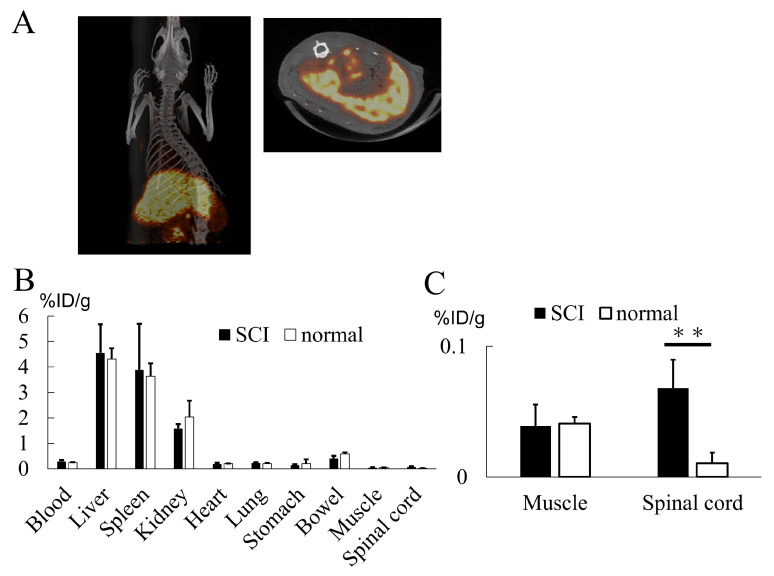
(**A**) ^64^Cu-labeled exosomes primarily localized to the liver and spleen. (**B**) Organ radioactivity analysis indicated substantial exosome accumulation in the liver, spleen, and kidney, with a limited presence of exosomes in the spinal cord. (**C**) Exosome accumulation in the spinal cord was significantly greater in the SCI model when compared to normal animals. Note that the Y-axis values in this graph are considerably smaller in comparison to those in (**B**) (*p* = 0.0234). ** *p* < 0.01.

## Data Availability

The data that support the findings of this study are available from the corresponding author [M.K.] upon reasonable request.

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
