# Peer review of "Intravenous Administration of Mesenchymal Stem Cell-Derived Exosome Alleviates Spinal Cord Injury by Regulating Neutrophil Extracellular Trap Formation through Exosomal miR-125a-3p"

_ijms, 2024, doi:10.3390/ijms25042406_

Round 1
Reviewer 1 Report
Comments and Suggestions for Authors
The study titled "Intravenous administration of mesenchymal stem cell-derived exosome alleviates spinal cord injury by regulating NETs formation through exosomal miR-125a-3p" investigates the therapeutic potential of mesenchymal stem cell (MSC)-derived exosomes in treating spinal cord injuries (SCI).
The study utilized a rat model of SCI, where injuries were induced and then treated with MSC-derived exosomes. Rats treated with MSC-derived exosomes showed significant improvements in motor function and a reduction in SCI size. There was also a notable decrease in neutrophil infiltration and NETs formation​​.
In vitro studies revealed that exosomes accumulate around the nucleus of activated neutrophils, and the presence of miR-125a-3p in exosomes played a crucial role in reducing NETs formation​​.
MSC-derived exosomes, particularly through their content of miR-125a-3p, play a significant role in alleviating spinal cord injury, largely by deactivating neutrophil NETs formation and concentrating in areas like the liver and spleen​​.
This study is conceptually valid, but there are several flaws:
Major revisions
· References to be redone: References 1 to 24 are correct, although I suggest checking reference 13. However, everything after reference 24 is incorrect.
· Redo Western Blot of CD63: Given the importance of characterizing exosomes, it would be advisable to have a better quality WB.
· Longitudinal Sections: It is necessary to have images of the longitudinal section for a more comprehensive view of the lesion.
· Figure 2: Low quality of the image.
· MPO Dosage: The dosage of MPO does not appear clear; To confirm this result, we recommend using a specific kit to quantify MPO in tissue and serum
· Confirm through Western Blot that NF-κB signaling pathway is targeted by miR-125a-3p.
· Questions: Why choose only females in the SCI model? Females have issues related to estrogens. Please provide a rationale.
· Questions: It has been demonstrated that the majority of exosomes are directed to the liver. Does this reduce their activity at the lesion site?
Minor revisions
· Punctuation.
· Line 317: “1:00” needs correction.
· In sub-paragraph 4.2.6, there is a piece of template.
Comments on the Quality of English Language
The quality of the English language can be improved
Author Response
We have carefully addressed the reviewer’s comment in our revised manuscript. The main corrections and point-by-point responses to the reviewer’s comments are provided below in the blue font, with all corresponding changes marked in red font in the manuscript.
Major revisions
- References to be redone: References 1 to 24 are correct, although I suggest checking reference 13. However, everything after reference 24 is incorrect.
We are so sorry to bother you with this type of trouble. It seems that references are mixed up when transferring manuscript to the IJMS template. We have re-checked the manuscript, and ref 13 is also removed.
- Redo Western Blot of CD63: Given the importance of characterizing exosomes, it would be advisable to have a better quality WB.
We changed the CD63 figure to better quality one. However, as you know, CD63 has 30-60 KDa of length, and the western blotting result is often seen vague.
- Longitudinal Sections: It is necessary to have images of the longitudinal section for a more comprehensive view of the lesion.
We thank the reviewer’s constructive opinion. We changed the figure for more comprehensive view of the lesion in Figure 2
- Figure 2: Low quality of the image.
- MPO Dosage: The dosage of MPO does not appear clear; To confirm this result, we recommend using a specific kit to quantify MPO in tissue and serum
Thank you for pointing important advice. We changed the figure for better understanding. MPO was stained by DAB staining and counting was done by automated cell counter. We believe that these methods are well validated, and adding specific kit will not add additional data. Therefore, the authors respectfully declined to re-evaluate the MPO.
- Confirm through Western Blot that NF-κB signaling pathway is targeted by miR-125a-3p.
Due to the limited time course for our revision (within 10 days), we were not able to complete the Western blotting. In that circumstance, we added the following sentences and references to prove the NF-kb signaling attenuated by miR-125. Following sentences are added in the discussion section.
the miR-125a-3p inhibitor successfully counteracts the NETs-suppressing effect of exosomes, confirming its role in anti-inflammatory processes. These findings are consistent with the work of Jin et al., who have exclusively reported that miR-125a-3p suppresses NETs by targeting the NF-κB signaling pathway mediated by Toll-like receptor 4 (TLR4)[32]. Kim et al. reported that mir-125a inhibits the NF-kb regulator tumor necrosis factor alpha-induced protein 3 to negatively affect NF-kb activity[33]. Additionally, mir125b, TLR4/JNK signaling has been reported to play a role in NETs formation by neutrophils[34]. MiR-125a-3p is also implicated in altering the phenotype of microglia by interfering with interferon regulatory factor 5 (IRF5). Knockdown of miR-125a-3p in MSC-derived exosomes resulted in increased IRF5 expression in spinal cord tissues[35]. Mir-125 is reported to show significant correlation with inflammatory bowel disease in human[36].
- Questions: Why choose only females in the SCI model? Females have issues related to estrogens. Please provide a rationale.
We thank the reviewer for pointing out the important question. Since male rats has longer urinary tract than female rats, it is more likely to present sepsis after spinal cord injury. We added the following sentence in the manuscript in method section.
Female rats were selected because they possess shorter urinary tract than male rats, which lowers the sepsis caused by anuresis after spinal cord injury.
- Questions: It has been demonstrated that the majority of exosomes are directed to the liver. Does this reduce their activity at the lesion site?
Thank you. In this article, the degradation of exosome is out of focus and we do not have definite answer for your question. However, our previous reports of intravenous MSC administration for spinal cord injury also showed similar result of liver accumulation. Due to the nature of drug degradation capacity of exosome, we think that exosome entrapped in the Kupffer cell are degraded. We added the following sentences in the discussion section.
The exact role of exosome accumulation in the liver remains uncertain. It could potentially be attributed to the uptake by Kupffer cells of eliminating unidentified substances.
Minor revisions
- Punctuation.
- Line 317: “1:00” needs correction.
- In sub-paragraph 4.2.6, there is a piece of template.
Thank you. We corrected the mistakes.
Reviewer 2 Report
Comments and Suggestions for Authors
In this manuscript, the authors propose a therapeutic approach for spinal cord injury utilizing injections of mesenchymal stem cell-derived exosomes. This method alleviates spinal cord injury by regulating NETs formation through exosomal miR-125a-3p. I think this is an interesting and valuable manuscript. Publishing is recommended after the authors address the following issues:
1. When using exosomes for in vivo experiments, whether the mass of exosomes or exosomal proteins is utilized for quantification.
2. Do you think that if we change the injection method, there will be better therapeutic outcomes? For example, directly injecting into the spinal cord parenchyma. This could be more challenging.
3. Some minor mistakes need to be double-checked, such as “scale bar =” in line 114, “?” in line 117. Lack of bar in Fig 4C. References need to be cited in a consistent format, e.g., in the intro section and the discussion section the numbers in lines 179 and 182.
Author Response
We have carefully addressed the reviewer’s comment in our revised manuscript. The main corrections and point-by-point responses to the reviewer’s comments are provided below in the blue font, with all corresponding changes marked in red font in the manuscript.
Reviewer 2
In this manuscript, the authors propose a therapeutic approach for spinal cord injury utilizing injections of mesenchymal stem cell-derived exosomes. This method alleviates spinal cord injury by regulating NETs formation through exosomal miR-125a-3p. I think this is an interesting and valuable manuscript. Publishing is recommended after the authors address the following issues:
- When using exosomes for in vivo experiments, whether the mass of exosomes or exosomal proteins is utilized for quantification.
Thank you for your valuable question. Exosome are quantified using BCA protein assay kit, and administered 100ug per animal. Quantifications were added to the method section
Protein concentration of exosome diluted in PBS was evaluated by Pierce BCA Protein Assay kit (23227, Thermo Fisher Scientific). Subsequently, 1 mL of AMSC-derived exosomes (100 μg) adjusted with PBS, or 1 mL of PBS alone, was administered via tail vein. Exosomes were given three consecutive days.
- Do you think that if we change the injection method, there will be better therapeutic outcomes? For example, directly injecting into the spinal cord parenchyma. This could be more challenging.
We thank the reviewer for raising critical comment. Discussing the different injection method is important for clinical setting. We added the following sentences in the discussion section.
In this report, we utilized intravenous transplantation as one of the most commonly used methods. However, other transplantation routes have also been reported for exosome administration against spinal cord injury. Intraspinal injection offers the possibility of delivering a high concentration of exosomes, but the injection itself carries the risk of additional spinal cord damage[40, 41]. Intranasal injection is currently gaining attention due to its minimally invasive method of delivering exosomes to the damaged spinal cord, although the absorption rate may vary depending on the patient's condition[39, 42, 43]. Intrathecal injection can deliver a high amount of exosomes with minimal invasive procedure[44].
- Some minor mistakes need to be double-checked, such as “scale bar =” in line 114, “?” in line 117. Lack of bar in Fig 4C. References need to be cited in a consistent format, e.g., in the intro section and the discussion section the numbers in lines 179 and 182.
Thank you. We corrected the mistakes.
Round 2
Reviewer 1 Report
Comments and Suggestions for Authors
I thank the authors for the changes made. The manuscript can be accepted in this final form